# The association between diet and mental health and wellbeing in young adults within a biopsychosocial framework

**Verena Rossa-Roccor**[1,2]*, **Chris G. Richardson**[1,3], **Rachel A. Murphy**[1,4], **Anne M. Gadermann**[1,3,5]

**1** School of Population and Public Health, University of British Columbia, Vancouver, British Columbia, Canada, **2** Institute for Resources, Environment and Sustainability, University of British Columbia, Vancouver, British Columbia, Canada, **3** Centre for Health Evaluation and Outcome Sciences, Providence Health Care Research Institute, St. Paul's Hospital, Vancouver, British Columbia, Canada, **4** Cancer Control Research, BC Cancer, Vancouver, British Columbia, Canada, **5** Human Early Learning Partnership, School of Population and Public Health, University of British Columbia, Vancouver, British Columbia, Canada

* verena.rossa-roccor@ubc.ca

## Abstract

### Objective

Predominantly plant-based diets can co-benefit human physical health and the planet. Young adults appear to be on the forefront of the shift to plant-based diets. However, little is known about the relationship between plant-based diets and mental health in this population even though mental health disorders contribute substantially to the global burden of disease, particularly among this age group.

### Design

In this cross-sectional study we utilize a biopsychosocial framework to assess the association between dietary intake and mental health and wellbeing. Mental health was assessed using self-reported measures of anxiety (GAD-7), depression (PHQ-9) and quality of life (single-item). Dietary intake in the prior month was assessed using a dietary screener (DSQ) and participants were asked to self-identify a diet preference (e.g., vegan).

### Setting and participants

339 university undergraduate students.

### Results

A principal component analysis of dietary intake found three dominant dietary patterns (plant-based, animal-based, and 'junk foods'); 28.1% (n = 95) of participants self-identified as pescatarian, vegetarian, vegan, other. The association between dietary patterns, diet preference and mental health was assessed through regression analysis. After controlling for covariables, we found a significant positive association between the junk food component and depression (z-score $\beta$ = .21, $p \leq$ .001; adj. $R^2$ = .39) and anxiety (z-score $\beta$ = .14;

**Data Availability Statement:** All relevant data are within the paper and its Supporting information files.

**Funding:** The author(s) received no specific funding for this work.

**Competing interests:** I have read the journal's policy and the authors of this manuscript have the following competing interests:The authors declare that they have no conflict of interest. RAM has received funds as a consultant from Pharmavite and research funds from the International Life Sciences Institute, North America. Neither relationship is relevant to the work presented in this manuscript.

$p \leq .001$; adj. $R^2 = .32$) while no association was found between plant-based, animal-based or self-identified diet preference and the mental health measures.

## Conclusions

We did not find a negative association between predominantly plant-based diet patterns and mental health and wellbeing. It is important to consider dietary composition and to conceptualize diet as a health behaviour that is embedded in a biopsychosocial framework.

## Introduction

Holistic health fields of enquiry such as planetary health view diet as being embedded in a complex system of interrelations between individual, social, cultural, and environmental factors. In March 2019, the EAT Lancet Commission on Healthy Diets from Sustainable Food Systems put forward the first global benchmark diet capable of sustaining human and planetary health [1]. The recommendations herein include two sets of frameworks: on the one hand, they specify food intake ensuring human health and on the other hand, they suggest specific planetary boundaries for food production. More specifically, these 'win-win diets' highlight the co-benefits of plant- over animal-based foods. The authors state that "when viewed together as an integrated human health and environmental sustainability agenda, 'win-win' diets, that fall within the safe operating space for food systems, will help to achieve global human health and environmental sustainability goals" [1].

The authors of this landmark report were able to draw on extensive evidence on the benefit of planetary healthy, i.e., predominantly plant-based diets, on physical health. However, the potential mental health impacts of these diets remain largely unknown. This lack of evidence is surprising given that mental and behavioural disorders are the leading cause of years lived with disability worldwide [2]. Depression and anxiety are the two leading mental health disorders in terms of global disease burden: Depressive disorders account for 40% of disability-adjusted life years (DALYs), anxiety disorders account for 15% of DALYs caused by all mental and substance use disorders [2]. Lifetime prevalence rates range from 10 to 15% for depression [3] and average 17% for all anxiety disorders combined [4]. Furthermore, mental illness often develops into a chronic, lifelong health issue that can have profound and devastating effects on an individual's life trajectory by impacting and disrupting social functioning and capital [5], educational attainment [6], economic output [7], and overall quality of life (QoL) [8].

Approximately 75% of all mental illnesses have their onset before the age of 25 [9–11]. University students in particular are vulnerable for depression, anxiety, and substance use disorders [12] and mental health issues within this population are on the rise [13, 14]. This warrants interventions that support students in this developmental phase which offers potential for preventive and early intervention as it is accompanied by significant brain development with elevated neural plasticity [15, 16]. To meet this growing need, research in the area of nutritional psychiatry is now considering dietary interventions to prevent and treat mental illnesses [17]. These interventions have the potential to contribute to improved emotional functioning and long-term health, as the adoption of a healthy diet during this developmental period may contribute substantially to the prevention of (chronic) non-communicable diseases in later stages of life [18].

Young people are also particularly likely to adopt plant-based diets [19, 20]. Current estimates see approximately 7% of Canada's population self-identifying as vegetarian or vegan

(compared to only 2% in 2003)–with those under the age of 35 being three times more likely than older generations to identify as vegetarian or vegan while predictions see this number increasing rapidly [20]. The numbers of those who do not completely abstain from meat or other animal-based products but aim to substantially decrease their consumption, particularly of greenhouse gas and water-intense red meats and ruminant products, are even higher: According to recent consumer polls, 43% of Canadians are aiming to incorporate more plant-based foods into their diets [21] which is reflected in a constant decline of overall per capita meat consumption in Canada over the last three decades [22].

There are different definitions and conceptualizations of plant-based diets. One approach is to assess diet preference, i.e., someone identifies as vegetarian, vegan, or newer categories such as 'flexitarian', a term describing individuals who eat "primarily vegetarian with the occasional inclusion of meat or fish" [23]. Using this categorical definition of plant-based diets, preliminary findings ranged from vegetarians reporting significantly better mood and less anxiety and stress compared to non-vegetarians [24, 25] to vegetarians having higher odds of lifetime prevalence of depression, anxiety, and physical disorders compared to non-vegetarians [26–30]. However, findings show that self-report of diet preference (i.e., stating whether one identifies as vegan, vegetarian, etc.) says very little about actual diet pattern and quality [31]. Certain plant-based foods such as whole grains, vegetables, legumes, nuts, and fruits are indeed known to have health benefits while high intake of others such as refined grains, fried potatoes, sweets and desserts, or fruit juices are generally considered unhealthy [32]. Therefore, compositions of diet patterns and diet quality of those who describe themselves as vegetarians, vegans, pescatarians, etc. likely differ greatly between individuals and need to be assessed more carefully. Research on the association between diet and mental health utilizing composite dietary measures such as dietary patterns and diet quality indices also shows heterogenous results. However, the trend seems to point towards better mental health among those following predominantly plant-based diets, i.e., diets that are high in vegetable, fruit, whole grain intake with moderate intake of fish and worse mental health among those following a 'Western' diet high in animal and processed foods [33–35].

In this study, we conceptualized plant-based diets as diet patterns that consist mostly or exclusively of plant-based foods. We assessed diet through two approaches: using a categorical definition of plant-based diet asking respondents to self-identify according to diet preferences (no preference, pescatarian, vegetarian, vegan, other with open text entry option) and the diet pattern-based approach through dietary pattern analysis. We then compared both approaches in terms of their association with depression, anxiety, and QoL hypothesizing that diet patterns high in plant foods rather than diet preference would be negatively associated with the outcomes.

One limitation that all previous studies on this topic have in common is their narrow focus on a primarily biomedical understanding of the relationship between diet and mental health. Neither mental health nor dietary behaviours exist in a vacuum. As described in an extensive body of research, stress, stressful life events, body image, physical activity, sleep, and social support are all predictors for mental health and wellbeing outcomes [36–41]. Simultaneously, these factors are conceptually related to diet and therefore fulfill the criteria of presenting possible confounders in the relationship under investigation in this study [42–47]. Previous studies have not sufficiently considered these factors, particularly the social dimension of dietary habits, in their theoretical frameworks and statistical models.

With this study, we therefore sought to address several gaps in the literature. We assessed diet patterns in a population of undergraduate university students. We further examined whether plant-based diet pattern and self-reported preferences are associated with mental health (depression and anxiety) and wellbeing (QoL) in this population (for simplicity, we

refer to both as 'mental health' herein). Finally, we extend the understanding of this relationship by considering this question within a biopsychosocial rather than a currently predominant biomedical framework in this field, thereby adding important confounding variables to the analysis.

# Methods

## Study design and participants

The study design was cross-sectional. We collected data through an online self-report survey from March to April of 2019. The main outcome variables of interest were depression, anxiety, and QoL as indicator of overall mental wellbeing. The main explanatory variable was diet as assessed through dietary pattern over the prior month as well as self-reported diet preference. The survey contained additional items on social support, health behaviours and status, body image, stress, stressful life events, and socioeconomic background.

We recruited participants among undergraduate students at the University of British Columbia (UBC), Vancouver, Canada through convenience sampling; data was collected anonymously. Excluding graduate students (n = 9) and cases that were missing items for any of the main outcome or main explanatory variables (n = 92) yielded a final analytic sample of n = 339 respondents.

## Measures

To assess dietary habits, we used the U.S. National Cancer Institute's Dietary Screening Questionnaire (DSQ) which asks about the frequency of consumption of select foods and beverages in the past 30 days. Evaluations have shown good agreement between estimates of intakes between the DSQ and multiple 24hr recalls with differences in means <2% and differences in prevalence <16% [48]. In its original version, the DSQ includes 26 items. The questionnaire was slightly altered in order to make it more appropriate for the local context and to include items that were relevant to this study such as consumption of poultry, additional dairy products, vegetarian meat alternatives, and non-dairy milk. The final version used in this study was not pilot-tested and had 28 items (see S1 File for complete questionnaire).

In addition to the DSQ, we included one item asking about dietary preference. Participants were asked if they identified as: a) pescatarian ('you eat fish, eggs, and dairy but no meat or poultry'); b) vegetarian ('you eat eggs and dairy but no fish, meat or poultry'); c) vegan ('you don't eat any animal products'); d) other ('please specify'; participants were given the option to enter text); e) none of the above.

We assessed QoL as a measure for overall mental wellbeing through a single-item measure ("In general, would you say your quality of life is. . .") with responses rated on a 5-point Likert scale (0 = poor, 1 = fair, 2 = good, 3 = very good, 4 = excellent). This single-item measure is one of the most widely used items to measure QoL and has been included in routinely used assessment tools such as the Patient-Reported Outcomes Measurement Information System Scale version 1.2 PROMIS® [49].

We assessed depressive symptoms using the 9-item Patient Health Questionnaire (PHQ-9) which is widely used in both clinical and research settings and has been validated for a variety of populations to detect and assess severity of depressive symptoms [50–53]. The total score ranges from 0 to 27. PHQ-9 scores of ≥10 have been reported to have a sensitivity of 88% and a specificity of 88% for major depression [53]. For clinical and diagnostic purposes, the measure can further be used to assess severity of symptoms applying cut-off scores. Cut-off scores for mild, moderate, moderately severe, and severe depression were found to be 5, 10, 15, and

20, respectively [53]. In general, a score ≥10 means that further clinical evaluation is indicated while a score ≥20 indicates that the individual may require psychotherapy and/or medication.

We assessed anxiety symptoms using the 7-item General Anxiety Disorder Questionnaire (GAD-7). Similar to the PHQ-9, this is a standard instrument to detect and assess the severity of anxiety disorder used widely for both clinical and research practices. Although originally designed to detect general anxiety disorder, it has been found that the GAD-7 is useful as a screening instrument for related anxiety disorders such as post-traumatic stress disorder, social anxiety disorder, and panic disorder [54]. The total score ranges from 0 to 21; for GAD-7 scores ≥10, sensitivity and specificity have been reported to be above 80% [55]. Much like the PHQ-9, the GAD-7 can further be used to assess severity of symptoms by applying cut-off scores. Cut-off scores for mild, moderate, and severe anxiety were found to be 5, 10, and 15, respectively [55]. In general, a score ≥10 means that further clinical evaluation is indicated while a score ≥15 indicates that the individual may require psychotherapy and/or medication.

## Statistical analyses and missing data

For descriptive purposes, we reported continuous variables through the mean and standard deviation (SD); for categorical variables, we reported frequencies.

The final sample consisted of n = 339 participants. In this analytic sample, the data on the main variables of interest (QoL, depression, anxiety, DSQ) was complete for all respondents. For covariables, responses such as 'prefer not to say' and 'don't know' were treated as missing data. Overall, missingness was fairly low in this sample. More specifically, missingness was as follows for the variables included in the logistic regression models: Age: 4.4%; Gender: 2.1%; Ethnicity: 3.2%; Sleep: 2.1%; Physical activity: 2.9%; Stressful life events: 3.5%; Weight satisfaction: 0.9%; Social support: 0.1%. Due to the sensitivity of some of these items as well as a high prevalence of 'prefer not to say' responses among the missing observations, we could not assume a missing completely at random pattern. Therefore, complete case analysis would not have been appropriate. Instead, multiple imputation is suggested as best practice [56]. We applied multiple imputation (Markov Chain Monte Carlo Method; five imputations) to address missing data for all covariables that were to be included in the multiple regression model based on the conceptual understanding of the relationship between diet and mental health in order to avoid underestimation of sampling error [57].

We applied principal component analysis (PCA) with varimax rotation as a data reduction approach for the evaluation of the DSQ items. The decision on how many components were to be retained was based on considering the combination of interpretability and conceptual reasoning of the emerging components, the eigenvalues (>1), the scree plot, and the percentage of variance explained by the components. Varimax rotation was chosen as it was assumed that emerging components would not be highly correlated with each other.

The PCA component scores for each participant were entered into regression models as the main explanatory variable when examining the relationship between diet pattern and mental health outcomes (using the total scores of the PHQ-9 and GAD-7 measures). We built three nested linear regression models per outcome using a hierarchical approach. We first entered sociodemographic factors (age, gender, ethnicity), then added lifestyle-related variables (physical activity, sleep, weight satisfaction, stress, stressful life events). The third step was to add social support as a known individual predictor for mental health. Finally, we added the main variables of interest (PCA component scores) to assess its additional contribution to the outcome of interest. These nested models were built for each outcome variable of interest: Model 1: QoL; Model 2: Depression; Model 3: Anxiety. This approach was repeated with self-reported diet preference as the main explanatory variable. Assumptions for linear models were met.

Assumptions were checked as follows: Independence of cases was given due to the study design (each observation exists only once, is not paired with an observation in another group nor is it influenced by another observation). Collinearity was assessed through VIF values (largest VIF should be <10; average VIF should not be substantially >1) and tolerance statistics (which should be >0.2; [58]). Normality was assessed through the normal probability plots of the residuals. Homoscedasticity and linearity were checked through residuals vs. fitted plots. All analyses were 2-tailed with a significance level of $p \leq 0.05$ and conducted with IBM SPSS Statistics 25[®].

## Results

### Sample characteristics and covariates

The total sample consists of n = 339 participants. Table 1 depicts detailed sample characteristics as well as frequencies of covariables that were included in the regression models such as health behaviours (physical activity and sleep), body image, overall stress, stressful life events, and social support. Overall, we found that almost none of the students (96.1%, n = 326) met the recommended amount of moderate or vigorous physical activity in the previous week. Three quarters of the participants (76.8%, n = 260) reported enough sleep to feel rested on a maximum of four days in the previous week. Two thirds of the students (66.6%, n = 226) experienced more than average or even tremendous stress over the 12 months preceding the survey. Approximately half of the students were somewhat, very, or extremely satisfied with their weight (52.6%, n = 178). Experiencing stressful life events that caused moderate or severe stress was reported by 76.3% (n = 259) of the students. Conversely, the majority of participants (80.4%, n = 272) reported having good, very good, or excellent satisfaction with their social relationships and activities.

### Diet

Three dietary components emerged from the PCA of the DSQ items. Component 1 (plant foods) was high in plant-based foods and non-animal-based dairy and meat alternatives as well as whole grains. Component 2 (animal foods) was high in animal-based foods such as different meats and dairy products. Component 3 (junk foods) was high in processed foods, snacks, and candies. The total variance explained by the retained three components was 40.6%. Details on loadings per component for each food item/group after varimax rotation can be seen in Table 2. For better interpretability, we removed food items/groups that did not load ≥0.4 on either of the components (namely, potatoes, tomato sauce, and fruit juice) from the final analysis [59]. In addition, loadings below 0.4 are omitted from the table to improve readability. Two items with cross-loadings were observed. The items on plant-based alternatives to meat and dairy milk had positive loadings on the first component (plant foods) and negative cross-loadings on the second component, which are the animal-based foods.

Almost one third of students (28.1%, n = 95) self-identified as either pescatarian, vegetarian, vegan or other (which were mostly on a spectrum of non-mainstream preferences such as reducetarian or flexitarian). See Table 3 for details.

### Mental health and wellbeing

As can be seen in Table 4, more than half of the participants (56.9%, n = 193) reported their overall QoL to be either very good or excellent with a mean score of 2.6 (±1.0) out of 5. The mean score for depression was 9.3 (±6.1) out of 27; the mean score for anxiety was 7.9 (±5.8) out of 21. In terms of clinical relevance, 75.0% (n = 254) of students had scores which indicate

**Table 1. Participant demographic and psychosocial characteristics.**

| Characteristic/Item | Item categories | mean | SD | n | % |
|---|---|---|---|---|---|
| †*Age* | | 19.5 | 1.9 | | |
| †*Gender identity* | Female | | | 224 | 66.1 |
| | Male | | | 109 | 32.1 |
| | Other (trans, queer, other) | | | 6 | 1.8 |
| *Sexual orientation* | Heterosexual | | | 257 | 75.8 |
| | Bisexual | | | 33 | 9.7 |
| | Gay/Lesbian | | | 6 | 1.8 |
| | Other | | | 28 | 8.3 |
| | Missing | | | 15 | 4.4 |
| *Relationship status* | Not in a relationship | | | 221 | 65.2 |
| | In a relationship | | | 95 | 28.0 |
| | Not sure | | | 12 | 3.5 |
| | Missing | | | 11 | 3.3 |
| †*Ethnicity* | White | | | 156 | 46.0 |
| | Asian | | | 135 | 39.8 |
| | Other | | | 48 | 14.2 |
| *Year in school* | 1st year | | | 211 | 62.2 |
| | 2nd year | | | 64 | 18.9 |
| | 3rd year | | | 28 | 8.3 |
| | 4th year | | | 19 | 5.6 |
| | Higher than 4th year undergrad | | | 9 | 2.7 |
| | Not seeking a degree | | | 1 | 0.3 |
| | Missing | | | 7 | 2.0 |
| *International student* | Yes | | | 120 | 35.4 |
| | No | | | 213 | 62.8 |
| | Missing | | | 6 | 1.8 |
| *Residence* | On-campus | | | 248 | 73.1 |
| | With parents | | | 34 | 10.0 |
| | Off-campus alone/with roommates/other | | | 48 | 14.2 |
| | Missing | | | 9 | 2.7 |
| †*Physical activity in the past 7 days* (20min of vigorous exercise or 30min of moderate exercise) | Never | | | 94 | 27.6 |
| | 1–3 days/week | | | 167 | 49.4 |
| | 4–6 days/week | | | 65 | 19.1 |
| | every day or more than once a day | | | 13 | 3.9 |
| †*Enough sleep to feel rested in the morning in the past 7 days* | $\leq$ 4 days/week | | | 260 | 76.8 |
| | $\geq$ 5 days/week | | | 79 | 23.2 |
| †*Weight satisfaction* | Not satisfied/slightly unsatisfied | | | 161 | 47.4 |
| | Somewhat satisfied | | | 101 | 29.9 |
| | Very/extremely satisfied | | | 77 | 22.7 |
| †*Perceived stress* | No/less than average stress | | | 25 | 7.4 |
| | Average stress | | | 88 | 26.0 |
| | More than average/tremendous stress | | | 226 | 66.6 |
| †*Stressful life events* | Mild stressors | | | 81 | 23.7 |
| | Moderate stressors | | | 149 | 44.1 |
| | Severe stressors | | | 109 | 32.2 |
| †*Social support* | Poor/fair | | | 67 | 19.6 |

*(Continued)*

**Table 1.** (Continued)

| Characteristic/Item | Item categories | mean | SD | n | % |
|---|---|---|---|---|---|
| | Good/very good/excellent | | | 272 | 80.4 |

[†] Data for these items was imputed.

the need for further evaluation concerning symptoms of depression; for anxiety 65.1% (n = 221) had scores indicating need for further evaluation. Of those who scored above 10 points for depression (n = 142), 16.2% (n = 23) would likely benefit from psychotherapy and/or medication; for those who scored above 10 points for anxiety (n = 110) this proportion is even higher with 48.2% (n = 53).

## Association between diet and mental health and wellbeing

Explanatory variables age, sleep, physical activity, stress, stressful life events, weight satisfaction, social support, and PCA scores were entered in the model as continuous variables. Gender and ethnicity were entered as categorical variables. The unadjusted linear regression analysis shows a significant association between several variables. The plant food dietary

**Table 2. Principal component analysis of dietary components and component loadings for dietary patterns after varimax rotation.**

| Food item/group | Component 1 (plant foods) | Component 2 (animal foods) | Component 3 (junk foods) |
|---|---|---|---|
| *Brown rice and whole grains* | 0.70 | | |
| *Beans and legumes* | 0.68 | | |
| *Nuts and seeds* | 0.66 | | |
| *Green leafy vegetables* | 0.66 | | |
| *Other vegetables* | 0.64 | | |
| *Fruit* | 0.63 | | |
| *Vegetarian/vegan meat alternatives* | 0.53 | -0.46 | |
| *Non-dairy milk* | 0.51 | -0.41 | |
| *Whole grain bread* | 0.49 | | |
| *Cereal* | 0.43 | | |
| *Poultry* | | 0.80 | |
| *Red meat* | | 0.75 | |
| *Processed meat* | | 0.68 | |
| *Fish and seafood* | | 0.61 | |
| *Cheese* | | 0.56 | |
| *Yoghurt* | | 0.50 | |
| *Dairy milk* | | 0.49 | |
| *Cookies, cake, pie* | | | 0.65 |
| *Ice cream* | | | 0.61 |
| *Donuts etc.*[†] | | | 0.60 |
| *Chocolate and candy* | | | 0.60 |
| *Soda* | | | 0.54 |
| *Pizza* | | | 0.51 |
| *Fried potatoes* | | | 0.50 |
| *Coffee or tea with sugar* | | | 0.41 |

[†]This item on the questionnaire included donuts, sweet rolls, Danish, muffins, pan dulce, and pop-tarts.

**Table 3. Self-reported diet preference.**

| Diet preference | n | % |
|---|---|---|
| *Pescatarian* | 13 | 4.0 |
| *Vegetarian* | 19 | 5.5 |
| *Vegan* | 37 | 10.8 |
| *Other* | 26 | 7.8 |
| *Do not identify as any of the above* | 244 | 71.9 |
| *Total* | 339 | 100 |

component was positively associated with QoL ($\beta$ = .20, $p \leq$ .001). The junk food component was positively associated with depression ($\beta$ = .26, $p \leq$ .001), while the animal food component and the plant food component were negatively associated with depression ($\beta$ = -.07, $p \leq$ .05 and $\beta$ = -.10, $p \leq$ .001, respectively). The junk food component was further positively associated with anxiety ($\beta$ = .18, $p$ = .001) and the animal food component was negatively associated with anxiety ($\beta$ = -.09, $p \leq$ .001). Table 5 shows the detailed results for the unadjusted associations between the main explanatory variable of interest (dietary pattern) and QoL, depression, and anxiety, respectively.

After adjusting for all covariables, the positive associations between the junk food component and depression and anxiety remain significant.

Model 1 (dietary pattern and QoL): After adjusting for all covariables, statistically significant negative associations were found between Asian ethnicity, stress, and QoL; significant positive associations were found for physical activity, weight satisfaction, and social support with QoL. Social support showed the strongest positive association for QoL ($\beta$ = .51 increase in QoL score; $p \leq$ .001).

**Table 4. Mental health and wellbeing.**

| Mental health item | Item categories | mean | SD | n | % |
|---|---|---|---|---|---|
| *QoL continuous (0 to 5)* | | 2.6 | 1.0 | | |
| *QoL ordinal* | Poor | | | 10 | 2.9 |
| | Fair | | | 31 | 9.1 |
| | Good | | | 105 | 31.1 |
| | Very good | | | 138 | 40.7 |
| | Excellent | | | 55 | 16.2 |
| *Depression score (0 to 27)* | | 9.3 | 6.1 | | |
| *Depression severity* | No depression | | | 85 | 25.0 |
| | Mild depression[†] | | | 112 | 32.9 |
| | Moderate depression | | | 73 | 21.7 |
| | Moderately severe depression | | | 46 | 13.6 |
| | Severe depression[‡] | | | 23 | 6.8 |
| *Anxiety score (0 to 21)* | | 7.9 | 5.8 | | |
| *Anxiety severity* | No anxiety | | | 118 | 34.9 |
| | Mild anxiety[†] | | | 111 | 32.7 |
| | Moderate anxiety | | | 57 | 16.8 |
| | Severe anxiety[‡] | | | 53 | 15.6 |

Abbreviation: QoL, quality of life.

[†]Cut-off for further evaluation.

[‡]Psychotherapy and/or medication are indicated.

**Table 5. Unadjusted effects of principal component analysis (PCA) diet components on quality of life (QoL), depression, and anxiety.**

| | QoL | | | Depression | | | Anxiety | | |
|---|---|---|---|---|---|---|---|---|---|
| | Unstandardized Beta | SE Beta | Standardized Beta ($\beta$) | Unstandardized Beta | SE Beta | Standardized Beta ($\beta$) | Unstandardized Beta | SE Beta | Standardized Beta ($\beta$) |
| **PCA plant foods** | 0.20 | 0.02 | **0.20**$^{**}$ | -0.10 | 0.02 | **-0.10**$^{**}$ | -0.06 | 0.02 | -0.06 |
| **PCA animal foods** | 0.01 | 0.02 | 0.01 | -0.07 | 0.02 | **-0.07**$^{**}$ | -0.09 | 0.02 | **-0.09**$^{**}$ |
| **PCA junk foods** | -0.04 | 0.02 | -0.03 | 0.26 | 0.02 | **0.26**$^{**}$ | 0.18 | 0.02 | **0.18**$^{**}$ |

$^*$ $p \leq .05$.

$^{**}$ $p \leq .001$.

Model 2 (dietary pattern and depression): After adjusting for all covariables, statistically significant negative associations were found between sleep, weight satisfaction, and social support with depression; a statistically significant positive association was found for stress and the junk food dietary component ($\beta$ = .21 increase in depression score; $p \leq .001$; $\Delta$adj. $R^2$ = .04).

Model 3 (dietary pattern and anxiety): After adjusting for all covariables, statistically significant positive associations were found between female gender, stress, stressful life events, and the junk food dietary component ($\beta$ = .14 increase in anxiety score; $p$ = .002; $\Delta$ adj. $R^2$ = .01) with anxiety. Social support was significantly negatively associated with anxiety.

Table 6 shows the detailed results for the three hierarchical multiple linear regression models that examined the association between dietary patterns and mental wellbeing outcomes controlling for covariables that reflected a biopsychosocial understanding of the relationship. $\Delta$adj. $R^2$ for each hierarchical step are reported in the footnotes.

Diet preference was not significantly associated with any of the outcome variables. Results are available upon request for these statistically non-significant findings.

## Discussion

### Interpretation

The final adjusted regression models show that the junk food component score was positively associated with depression and anxiety while there were no significant associations between the plant food or the animal food component and any of the mental health outcomes. While the additional variance explained by the dietary component (junk food) with regard to the mental health outcomes seems small ($\Delta$ adjusted $R^2$ for the model with depression as outcome = .04 and = .01 for the model with anxiety as outcome, respectively), the magnitude of the standardized regression coefficient is comparable to other covariates in the model that are known to be strongly associated with mental health outcomes (e.g., social support). There are two possible explanations for this. In line with the understanding that mental health and diet exist within a biopsychosocial framework, food intake contributes to a complex network of variables that reduce or enhance the risk for adverse mental health outcomes such as social support and relationships. Second, it has been found that self-reported dietary data typically leads to an underestimation of associations [60]. The possibility of underestimation of the association is therefore likely in this study which would mean that the true effect size may be larger. The non-significant trend in the expected direction for the association of the plant food component with mental health outcomes after adjusting for the covariables in this study should thus be interpreted as inconclusive and needs further exploration.

**Table 6. Adjusted effects of principal component analysis (PCA) diet components on quality of life (QoL), depression, and anxiety.**

| | Model 1: QoL | | | Model 2: Depression | | | Model 3: Anxiety | | |
|---|---|---|---|---|---|---|---|---|---|
| | Unstandardized Beta | SE Beta | Standardized Beta ($\beta$) | Unstandardized Beta | SE Beta | Standardized Beta ($\beta$) | Unstandardized Beta | SE Beta | Standardized Beta ($\beta$) |
| *Step 1* | | | | | | | | | |
| Constant | 3.72 | 0.52 | 0.08 | 3.58 | 3.36 | -0.10 | 3.31 | 3.15 | -0.25 |
| Age | -0.05 | 0.03 | **-0.11*** | 0.26 | 0.17 | 0.09 | 0.16 | 0.16 | 0.05 |
| Female gender[†] | 0.20 | 0.12 | 0.21 | -0.10 | 0.71 | -0.02 | 1.31 | 0.67 | **0.23*** |
| Other gender[†] | -0.52 | 0.42 | -0.55 | 4.57 | 2.80 | 0.75 | 4.68 | 2.62 | 0.81 |
| Asian ethnicity[‡] | -0.50 | 0.11 | **-0.52**** | 1.34 | 0.72 | 0.22 | 0.79 | 0.68 | 0.14 |
| Other ethnicity[‡] | -0.02 | 0.16 | -0.02 | 0.73 | 1.02 | 0.12 | 1.40 | 0.96 | 0.24 |
| *Step 2* | | | | | | | | | |
| Constant | 3.55 | 0.54 | -0.03 | 5.62 | 3.13 | 0.05 | 0.85 | 3.00 | -0.12 |
| Age | -0.03 | 0.03 | -0.07 | 0.08 | 0.14 | 0.03 | 0.02 | 0.14 | 0.01 |
| Female gender[†] | 0.26 | 0.10 | **0.27*** | -0.84 | 0.60 | -0.14 | 0.63 | 0.58 | 0.11 |
| Other gender[†] | -0.11 | 0.40 | -0.11 | 1.10 | 2.36 | 0.18 | 1.49 | 2.26 | 0.26 |
| Asian ethnicity[‡] | -0.40 | 0.10 | **-0.41**** | 0.56 | 0.62 | 0.09 | 0.35 | 0.60 | 0.06 |
| Other ethnicity[‡] | 0.07 | 0.14 | 0.07 | 0.04 | 0.86 | 0.01 | 0.76 | 0.82 | 0.13 |
| Sleep | 0.04 | 0.02 | 0.09 | -0.58 | 0.15 | **-0.20**** | -0.32 | 0.14 | **-0.12*** |
| Physical activity | 0.07 | 0.02 | **0.14*** | -0.36 | 0.15 | **-0.12*** | -0.19 | 0.14 | -0.07 |
| Stress | -0.30 | 0.06 | **-0.27**** | 2.16 | 0.34 | **0.31**** | 2.55 | 0.33 | **0.39**** |
| Stressful life events | 0.02 | 0.07 | 0.02 | 0.72 | 0.40 | 0.09 | 1.02 | 0.39 | **0.13*** |
| Weight satisfaction | 0.11 | 0.04 | **0.12*** | -1.15 | 0.27 | **-0.20**** | -0.54 | 0.26 | **-0.10*** |
| *Step 3* | | | | | | | | | |
| Constant | 1.74 | 0.48 | 0.06 | 10.54 | 3.24 | 0.01 | 4.43 | 3.13 | -0.15 |
| Age | -0.01 | 0.02 | -0.01 | -0.01 | 0.14 | -0.01 | -0.04 | 0.14 | -0.02 |
| Female gender[†] | 0.06 | 0.09 | 0.06 | -0.30 | 0.60 | -0.05 | 1.02 | 0.58 | 0.18 |
| Other gender[†] | -0.22 | 0.33 | -0.23 | 1.40 | 2.30 | 0.23 | 1.70 | 2.23 | 0.30 |
| Asian ethnicity[‡] | -0.28 | 0.09 | **-0.30*** | 0.26 | 0.60 | 0.04 | 0.13 | 0.59 | 0.02 |
| Other ethnicity[‡] | 0.11 | 0.12 | 0.11 | -0.06 | 0.83 | -0.01 | 0.69 | 0.81 | 0.12 |
| Sleep | 0.01 | 0.02 | 0.03 | -0.50 | 0.14 | **-0.17**** | -0.19 | 0.14 | -0.09 |
| Physical activity | 0.06 | 0.02 | **0.13*** | -0.35 | 0.14 | **-0.11*** | -0.19 | 0.14 | -0.07 |
| Stress | -0.17 | 0.05 | **-0.16**** | 1.84 | 0.34 | **0.27**** | 2.31 | 0.33 | **0.36**** |
| Stressful life events | 0.05 | 0.06 | 0.04 | 0.65 | 0.39 | 0.08 | 0.96 | 0.38 | **0.13*** |
| Weight satisfaction | 0.09 | 0.04 | **0.10*** | -1.10 | 0.26 | **-0.19**** | -0.50 | 0.26 | **-0.10*** |
| Social support | 0.46 | 0.04 | **0.51**** | -1.26 | 0.28 | **-0.22**** | -0.92 | 0.27 | **-0.17**** |
| *Step 4* | | | | | | | | | |
| Constant | 1.77 | 0.49 | 0.07 | 10.30 | 3.22 | -0.02 | 4.20 | 3.15 | -0.17 |

*(Continued)*

**Table 6.** (Continued)

| | Model 1: QoL | | | Model 2: Depression | | | Model 3: Anxiety | | |
|---|---|---|---|---|---|---|---|---|---|
| | Unstandardized Beta | SE Beta | Standardized Beta (β) | Unstandardized Beta | SE Beta | Standardized Beta (β) | Unstandardized Beta | SE Beta | Standardized Beta (β) |
| Age | -0.01 | 0.02 | -0.01 | 0.01 | 0.14 | 0.01 | -0.04 | 0.14 | -0.01 |
| Female gender[†] | 0.04 | 0.10 | 0.04 | -0.02 | 0.63 | -0.01 | 1.28 | 0.62 | **0.22**[*] |
| Other gender[†] | -0.23 | 0.34 | -0.24 | 1.41 | 2.26 | 0.23 | 1.73 | 2.22 | 0.30 |
| Asian ethnicity[‡] | -0.28 | 0.09 | **-0.29**[*] | 0.23 | 0.60 | 0.04 | 0.08 | 0.60 | 0.01 |
| Other ethnicity[‡] | 0.11 | 0.12 | 0.12 | 0.02 | 0.82 | 0.01 | 0.72 | 0.80 | 0.12 |
| Sleep | 0.01 | 0.02 | 0.02 | -0.49 | 0.14 | **-0.17**[**] | -0.25 | 0.14 | -0.09 |
| Physical activity | 0.06 | 0.02 | **0.12**[*] | -0.25 | 0.15 | -0.08 | -0.10 | 0.15 | -0.04 |
| Stress | -0.17 | 0.05 | **-0.16**[**] | 1.82 | 0.33 | **0.27**[**] | 2.30 | 0.33 | **0.36**[**] |
| Stressful life events | 0.05 | 0.06 | 0.04 | 0.42 | 0.38 | 0.05 | 0.81 | 0.39 | **0.11**[*] |
| Weight satisfaction | 0.09 | 0.04 | **0.10**[*] | -0.96 | 0.26 | **-0.17**[**] | -0.42 | 0.25 | -0.08 |
| Social support | 0.46 | 0.04 | **0.51**[**] | -1.36 | 0.28 | **-0.23**[**] | -0.97 | 0.27 | **-0.18**[**] |
| PCA plant foods | 0.04 | 0.04 | 0.05 | -0.07 | 0.30 | -0.01 | -0.19 | 0.29 | -0.03 |
| PCA animal foods | 0.01 | 0.04 | 0.01 | -0.15 | 0.28 | -0.02 | -0.14 | 0.28 | -0.02 |
| PCA junk foods | -0.01 | 0.04 | -0.01 | 1.26 | 0.27 | **0.21**[**] | 0.83 | 0.27 | **0.14**[*] |

[†]Reference category: Male gender.

[‡]Reference category: Caucasian ethnicity.

[*]p≤.05.

[**] p≤.001.

Note Model 1: Adjusted $R^2$ = .08 for Step 1; Δ adj $R^2$ = .13 for Step 2; Δ adj $R^2$ = .23 for Step 3; Δ adj $R^2$ = .00 for Step 4.

Note Model 2: Adjusted $R^2$ = .01 for Step 1; Δ adj $R^2$ = .30 for Step 2; Δ adj $R^2$ = .04 for Step 3; Δ adj $R^2$ = .04 for Step 4.

Note Model 3: Adjusted $R^2$ = .02 for Step 1; Δ adj $R^2$ = .27 for Step 2; Δ adj $R^2$ = .02 for Step 3; Δ adj $R^2$ = .01 for Step 4.

By adjusting for important confounders which have not previously been included in studies of mental health and diet, this present study corroborates the finding that 'unhealthy' dietary patterns are associated with depression and anxiety [34, 61, 62]. One possible causal pathway through which unhealthy foods such as processed (i.e., "foods that are altered to add or introduce substances that substantially change their nature or use") and ultra-processed foods (i.e., "industrial formulations, usually made mainly or solely from industrial ingredients, which contain little or no whole food") [63] may negatively impact mental health is that of inflammatory reactions and oxidative stress [64, 65]. Interestingly, in this study, processed plant-based foods such as meat replacements did not load strongly on the junk food component but actually showed high component loadings in the plant food component. However, future research is needed to understand whether these foods reflect healthier dietary patterns, particularly given the rise of consumer demand for plant-based processed foods. For example, the development and application of a dietary screening measure that captures these foods in more detail may render a better understanding of these emerging dietary patterns.

The prevalence of clinically-relevant levels of depression and anxiety is high in this sample of 339 undergraduate students. These prevalence rates are in line with findings from previous research on the mental health of students indicating that the prevalence of mental health issues is higher among university students than in the general population [12, 66]. There are several hypotheses why this may be the case. The typical age-of-onset of many psychiatric disorders overlaps with entry into university [11]; and the transition into university presents a stressful life event which is accompanied by homesickness, potentially social isolation, financial burden and pressure, and stress—all of which are risk factors for the development of depression and anxiety [67].

Conversely, more than half of the participants also report their overall QoL to be either very good or excellent. While this may at first seem counterintuitive, this is actually in line with the concept of QoL being a measure of a full continuum of (mental) wellbeing wherein the presence of symptoms of a disorder such as depression and anxiety merely present one dimension. It has been found, for example, that factors such as self-esteem or social support mitigate the role of depressive symptoms on QoL [68]. Fahy et al. also found that the strongest predictors for QoL in people with severe mental illness were unmet basic, social, and functional needs (in combination with symptom severity) [69]. Thus, assessing QoL in addition to screening for depression and anxiety provides a more complete picture of mental wellbeing and its associated factors in this study.

The possibility of reverse causality is another important consideration that researchers have identified [30] with one prospective cohort study providing probable evidence for reverse causality between depression and a healthy diet pattern [70]. Because dietary changes are perceived as a means to shape health, it can be hypothesized that a change in dietary behaviour could follow the onset of mental health issues as a form of 'self-medication'. Conversely, the 'self-medication' may also take on the form of an unhealthy diet consistent of foods high in sugar and fat to feel instant gratification [71].

## Strengths and limitations

This study utilizes a biopsychosocial conceptual understanding of the relationship between exposure and outcome and the inclusion of confounding variables that goes beyond a narrow biomedical approach. The present study builds on previous studies on this topic. Nevertheless, in order to eliminate temporal ambiguity, confounding, and response biases, more sophisticated study designs are needed in future investigations.

The relatively small sample size and the potential lack of power means that some effects of the explanatory variables may have remained uncovered in this study and that an underestimation for these effects was likely present. This could be mitigated in future studies with larger sample sizes. This may have been amplified by the finding that self-reported data on diet typically leads to an underestimation of associations [60].

In this present study, all participants were undergraduate students. The external validity of this study beyond the student population is thus limited as university students differ from their non-student peers and the general population in several characteristics, e.g., in terms of socioeconomic backgrounds, lifestyle behaviours, or substance use [72, 73]. In relation to the general population of undergraduates at the university, this sample was, however, fairly representative as its sociodemographic composition was comparable with that of the overall undergraduate student population. It is also important to note that across the continuum of depressive and anxiety symptoms, eating behaviours may differ (e.g., individuals with major depressive disorder often suffer from very reduced appetite and their overall food intake may be severely decreased). Thus, the findings of this study may not apply to individuals suffering

from major depressive and anxiety disorders as this study was not conducted on a clinical sample.

All collected information was exclusively self-reported which introduces non-response, reporting, and recall biases. In this study, the primary interest was to assess diet patterns rather than exact nutrient intake. Self-reported diet data have been deemed adequate and superior to non-self-reported measures such as biomarkers especially when analyzing diet patterns as they provide more complete information on the composition of the overall diet [60]. Given that dietary screeners are less burdensome on participants than methods like repeated 24-hr recalls while still providing sufficient information on food intake, we chose to use the DSQ as measure for diet [48]. Its limitations include that it does not allow for conclusions about the actual amount of food intake nor does it capture the full range of foods in one's diet. We further used a slightly amended version of the DSQ to make it more applicable to the local context. These amendments were minor and consisted of replacement of foods with limited relevance with those that were more relevant to the research question. We did not conduct a pilot study with this altered measure. However, the wording and structure of the questions were not altered, we therefore assume the potential bias due to issues of interpretability to be negligible. To mitigate the subjectivity and biased information from self-reported mental health issues, this study included validated screening instruments (one-item QoL scale, PHQ-9, GAD-7). Single-item measures for capturing covariates were chosen to reduce respondent burden. While these measures may not be as detailed as multi-dimensional measures, they have been shown to be useful when measuring aspects of respondents' health [74]. While answers are still self-reported, all measures used herein have been extensively validated.

It is also important to note that the different measures in this study assessed variables with different time frames. More specifically, the DSQ asked about food intake within the past 30 months whereas the PHQ-9 and GAD-7 assessed symptoms in the past two weeks. Other items evaluating covariables did not consider a specific time frame. Hence, based on the time-frames of the measures and the cross-sectional study design, inferences can only be made about the prevalence of the exposures and the outcomes and their degree of association at one point in time.

## Conclusions and future directions

We show that plant foods are positively associated with mental health outcomes but this association is attenuated after adjusting for other variables in our biopsychosocial model. We further found no relationship between categories of certain diet preferences such as vegetarian or vegan and mental health. These findings support approaches in nutritional epidemiology that employ dietary pattern analyses. By taking a more sophisticated approach to covariate selection and dietary assessment, our findings add to the evidence that contrasts a widely accepted, albeit outdated, perception of vegetarians and vegans as unhealthy individuals at risk for nutrient deficiency [75, 76]. This study provides a preliminary indication that the 'win-win' situation for planetary and somatic health of predominantly plant-based diets is not a 'win-win-lose' situation for mental health. Further research will be needed to confirm or refute this finding and would benefit from the inclusion of socioeconomic and cultural determinants as additional covariates of interest. For example, the issue of food security greatly impacts one's ability to access healthy foods and has been associated with major depressive disorder in US women [77]. In addition, the ability to procure culturally-appropriate foods, which has been nearly eliminated by a colonial food system, is an issue of great extent for Indigenous communities and food traditions across the globe. How this may interact with mental health is of great importance and has been neglected in the public health literature at this point. Moreover, most

of the studies on this topic have thus far have been conducted in North America, Europe, or Australia. Insights from countries and cultures other than Western nations would be helpful in understanding cross-cultural differences. Integrating research from social sciences, community action and participatory research, and findings from qualitative studies would also play a pivotal role in understanding the complex relationships under investigation.

## Supporting information

**S1 File. Questionnaire.**
(PDF)

## Acknowledgments

We are grateful for the support of David Gill and the University of British Columbia's SEEDS program who guided this project in an encouraging way and substantially facilitated the integrated knowledge translation and community-based approach of this research. We would also like to extend our gratitude to the members of the stakeholder group, most notably Melissa Baker-Wilson and David Speight for their enthusiasm, support, and open-mindedness. It was a true pleasure to be surrounded by these like-minded, visionary change makers.

## Author Contributions

**Conceptualization:** Verena Rossa-Roccor, Chris G. Richardson, Rachel A. Murphy, Anne M. Gadermann.

**Data curation:** Verena Rossa-Roccor.

**Formal analysis:** Verena Rossa-Roccor.

**Methodology:** Verena Rossa-Roccor, Chris G. Richardson, Anne M. Gadermann.

**Supervision:** Chris G. Richardson, Rachel A. Murphy, Anne M. Gadermann.

**Writing – original draft:** Verena Rossa-Roccor.

**Writing – review & editing:** Chris G. Richardson, Rachel A. Murphy, Anne M. Gadermann.

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
