## [Decision Letter · Decision Letter 0]

18 Feb 2021

PONE-D-20-39899

The association between diet and mental health and wellbeing in young adults within a biopsychosocial framework

PLOS ONE

Dear Dr. Rossa-Roccor,

Thank you for submitting your manuscript to PLOS ONE. After careful consideration, we feel that it has merit but does not fully meet PLOS ONE’s publication criteria as it currently stands. Therefore, we invite you to submit a revised version of the manuscript that addresses the points raised during the review process.

I am very much thankful to the reviewers for their deep and thorough review. The article requires a major revision by referring to the reviewer's comments. Statistical analysis should be done appropriately and meticulously.

We look forward to receiving your revised manuscript.

Kind regards,

Sıdıka Bulduk, Prof. Dr.

Academic Editor

PLOS ONE

Journal Requirements:

2.Thank you for submitting the above manuscript to PLOS ONE. During our internal evaluation of the manuscript, we found significant text overlap between your submission and your master's thesis (https://open.library.ubc.ca/cIRcle/collections/ubctheses/24/items/1.0379616).

Please note that there are copyright implications to reusing this text, as the thesis is licensed under the CC-BY-NC-ND 4.0 license (https://creativecommons.org/licenses/by-nc-nd/4.0/), which does not allow for commercial reuse without permission from the copyright holder. In order to publish any previously copyrighted material, PLOS ONE requires permission from the original copyright holder of the content to publish it under the CC BY 4.0 license.

Please clarify whether the authors have received written permission from University of British Columbia to publish this content specifically under the CC BY 4.0 license and upload the granted permission to the manuscript as a supporting information file.

In your revision, please either revise the duplicated text or provide specific permission from the copyright holder to republish the content of your master's thesis under a CC BY 4.0 license. Please note that further consideration is dependent on the submission of a manuscript that addresses these concerns about the overlap in text with published work.

3.We note that the grant information you provided in the ‘Funding Information’ and ‘Financial Disclosure’ sections do not match.

4.Thank you for stating the following in the Competing Interests section:

"I have read the journal's policy and the authors of this manuscript have the following competing interests:The authors declare that they have no conflict of interest. RAM has received funds as a consultant from Pharmavite and research funds from the International Life Sciences Institute, North America. Neither relationship is relevant to the work presented in this manuscript."

5. We noted in your submission details that a portion of your manuscript may have been presented or published elsewhere.

"This manuscript is based on the Master’s thesis of the corresponding author, Verena Rossa-Roccor. The thesis has been published at the UBC Theses and Dissertation Collection. A shortened version of the thesis has been published in the form of a report for stakeholders and knowledge users with the UBC food system and wellbeing operations in the UBC Sustainability Library."

Reviewers' comments:

Reviewer's Responses to Questions

**Comments to the Author**

1. Is the manuscript technically sound, and do the data support the conclusions?

Reviewer #1: Yes

Reviewer #2: No

Reviewer #3: Partly

2. Has the statistical analysis been performed appropriately and rigorously? 

Reviewer #1: Yes

Reviewer #2: Yes

Reviewer #3: No

3. Have the authors made all data underlying the findings in their manuscript fully available?

Reviewer #1: Yes

Reviewer #2: Yes

Reviewer #3: Yes

4. Is the manuscript presented in an intelligible fashion and written in standard English?

Reviewer #1: Yes

Reviewer #2: Yes

Reviewer #3: Yes

5. Review Comments to the Author

Reviewer #1: The manuscript has got an interesting topic and an appropriate methodology. It was written intelligibly in general. Also, the authors highlighted its limitations fairly. I believe that will contribute to the literature on this topic.

Reviewer #2: This manuscript is an intelligible. And it is wriitten standart English.After correcting, the article is suitable for publication.

Please see attached annotated PDF from Reviewer #2

Reviewer #3: The manuscript entitled ‘The association between diet and mental health and wellbeing in young adults within a biopsychosocial framework’ with the aim to study the relationship between plant-based diet and mental health among young adults.

The manuscript can be further improved based on the comments below.

Methods

Since few amendments were done on DSQ, the information on whether the questionnaire was tested/piloted prior to the study to be stated.

Statistical analyses and missing data

Line 210, information on percentage/pattern of missing data to be stated and discussed.

Results

Line 226, some measurements require proper tool/inventory to measure such as perceived stress, stressful life events, sleep. Using one or two questions may not enough to capture the real condition.

The type of data for the explanatory variables and how it was treated in the regression analysis to be clearly stated. A statement on statistical test assumptions fulfillment would be useful.

Table 1, the number of missing data for individual variable to be displayed.

Table 2, for component 2, the reason(s) for the two negative values to be explained.

Table 3, total N to be stated. Title too short.

Table 4, for item categories, the categories name to begin with capital letter.

Table 5, B to be written as Unstandardized B.

Line 281, for anxiety 65.1% but in Table 4 it showed 65.2%.

Line 297 – 303, results to be presented in table form.

Decimal point for percentages in the text in the results section to be standardized i.e. at least 1 decimal point.

Discussion

Line 401, for the sentence ‘relatively small sample size and the associated lack of power’ figures to be provided to support the statement.

Not all references are conformed to the journal format.

6. PLOS authors have the option to publish the peer review history of their article (what does this mean?). If published, this will include your full peer review and any attached files.

Reviewer #1: No

Reviewer #2: No

Reviewer #3: No

---

## [Author Response · Author response to Decision Letter 0]

23 Apr 2021

The line numbers refer to the revised manuscript with TRACKED changes.

Responses to Reviewer #1:

n/a

Responses to Reviewer #2:

Reviewer submitted comments in manuscript PDF as follows:

Line 57: The reviewer suggests to write ‘here’ instead of ‘herein’. We believe that ‘herein’ is the grammatically correct term to use in this case as we mean to say ‘in this text’ (see definition of ‘herein’: https://www.oxfordlearnersdictionaries.com/definition/english/herein) 

Line 145: The reviewer suggests to add ‘overall quality of life’ in addition to the abbreviation QoL. We would like to highlight that the definition for the abbreviation was already given in Line 80 (Introduction section) when the abbreviation is first mentioned in the text. This in accordance with the formatting guideline of the journal, see here: https://journals.plos.org/plosone/s/submission-guidelines

Line 243: The reviewer suggests to write (n=339) after sentence. We are unclear about this suggestion since the number of participants is already clearly stated in said sentence: “The total sample consists of n=339 participants.” Do you mean for us to state the total number of participants again in that same sentence or elsewhere?

Table 1 (Line 256): The reviewer notes that total count of observations for item ‘sexual orientation’ does not add up to n=339 and suggests we “correct all the tables”. Please note that, originally, we state in the footnote of the table, “n may vary due to missing data”. We did not impute missing data for these items. Therefore, for some variables, we have less than 339 observations. However, to respond appropriately to the reviewer’s concern and to make the tables more intuitive for the reader, we have added rows depicting the n of missing observations for each of the non-imputed items in Table 1. 

Table 4 (Line 289): The reviewer notes that percentages do not add up to 100%. This discrepancy happened due to rounding to one decimal. However, we have revised the tables and have rounded the decimals so that categories in tables now add up to 100%. We did find one typo which was corrected (see Table 4 QoL ordinal, Very good 4.7 instead of 4.1.). 

Responses to Reviewer #3:

Methods

Since few amendments were done on DSQ, the information on whether the questionnaire was tested/piloted prior to the study to be stated.

We changed the following sentence in the Method section in Line 163: “The final version used in this study was not pilot-tested and had 28 items (see supplementary materials for questionnaire)”. We further added the following to the Discussion section in Line 430: “We further used a slightly amended version of the DSQ to make it more applicable to the local context. These amendments were minor and consisted of replacement of foods with limited relevance with those that were more relevant to the research question. We did not conduct a pilot study with this altered measure. However, the wording and structure of the questions were not altered, we therefore assume the potential bias due to issues of interpretability to be negligible.”

Statistical analyses and missing data

Line 211, information on percentage/pattern of missing data to be stated and discussed.

We added the following to the revised manuscript (Line 204): “Overall, missingness was fairly low in this sample. More specifically, missingness was as follows for the variables included in the logistic regression models: Age: 4.4%; Gender: 2.1%; Ethnicity: 3.2%; Sleep: 2.1%; Physical activity: 2.9%; Stressful life events: 3.5%; Weight satisfaction: 0.9%; Social support: 0.1%. Due to the sensitivity of some of these items as well as a high prevalence of ‘prefer not to say’ responses among the missing observations, we could not assume a missing completely at random pattern. Therefore, complete case analysis would not have been appropriate. Instead, multiple imputation is suggested as best practice” [Sterne, J. A. C., White, I. R., Carlin, J. B., Spratt, M., Royston, P., Kenward, M. G., Wood, A. M., & Carpenter, J. R. (2009). Multiple imputation for missing data in epidemiological and clinical research: Potential and pitfalls. BMJ, 338(7713), 157-160. https://doi.org/10.1136/bmj.b2393]. 

Results

Some measurements require proper tool/inventory to measure such as perceived stress, stressful life events, sleep. Using one or two questions may not enough to capture the real condition.

We agree with the reviewer that some of the concepts used as covariables are complex and would indeed benefit from more extensive measurements. We decided to use single-item measures or more simple measures, respectively, for some of these concepts to keep respondent burden low. The questionnaire used in this study was already fairly long (since the measures for the outcome variable and the main explanatory variables were extensive) but we did not want to omit these important variables as their inclusion presents a significant advancement over previous studies in this area. To keep this trade-off as insignificant as possible, we exclusively used single-item measures that have been widely used and validated. Furthermore, it has been shown that single-item measures often suffice to measure concepts such as quality of life and others (Bowling, A. (2005). Just one question: If one question works, why ask several? Journal of Epidemiology and Community Health (1979), 59(5), 342-345. https://doi.org/10.1136/jech.2004.021204).

The items on overall quality of life and on social support were taken from the Patient-Reported Outcomes Measurement Information System Scale version 1.2 (PROMIS®). The PROMIS® is a measure to assess patient-reported health outcomes and previous research has shown evidence for its reliability and precision in measuring health-related symptoms and functioning (Cella, D., Riley, W., Stone, A., Rothrock, N., Reeve, B., Yount, S., … Hays, R. (2010). The Patient-Reported Outcomes Measurement Information System (PROMIS) developed and tested its first wave of adult self-reported health outcome item banks: 2005–2008. Journal of Clinical Epidemiology, 63(11), 1179–1194. https://doi.org/10.1016/j.jclinepi.2010.04.011).

Stressful life events were measured with the College Student’s Stressful Event Checklist. This checklist contains 32 items which had been modified from its original version for adults, the Social Readjustment Rating Scale to reflect appropriate events in the population of college students. Each item is assigned a specific value that corresponds to the potential stress magnitude of the event (Holmes, T. H., & Rahe, R. H. (1967). The social readjustment rating scale. Journal of Psychosomatic Research, 11(2), 213–218. https://doi.org/10.1016/0022-3999(67)90010-4). Values are summed up to calculate an overall score which reflects mild (total score <150), moderate (total score between 150 and 300) or severe stress (total score >300) due to these events. Despite its dated origin, this measure and its adapted versions continue to be among the most widely used and cited instruments to measure stressful life events and have been found to be a robust measure to identify events that may lead to stress-related outcomes (Scully, J. A., Tosi, H., & Banning, K. (2000). Life Event Checklists: revisiting the Social Readjustment Rating Scale after 30 years. Educational and Psychological Measurement, 60(6), 864–876. https://doi.org/10.1177/00131640021970952).

Items on overall stress, physical activity, sleep, satisfaction with one’s weight (as a proxy for body image), and sociodemographic variables were adapted from the National College Health Assessment II (NCHA-II) of the American College Health Association. The NCHA-II is a survey that collects data on student health status and behaviours as well as factors influencing academic performance in order to provide universities with information on students’ health needs and previous research has shown evidence for adequate reliability and validity of the measure (American College Health Association. (2013). American College Health Association-National College Health Assessment II: reliability and validity analyses 2011. Hanover, MD: American College Health Association.). 

We added the following to the Discussion section (see Line 437): “Single-item measures for capturing covariates were chosen to reduce respondent burden. While these measures may not be as detailed as multi-dimensional measures, they have been shown to be useful when measuring aspects of respondents’ health.”; and we added this reference to the list: Bowling, A. (2005). Just one question: If one question works, why ask several? Journal of Epidemiology and Community Health (1979), 59(5), 342-345. https://doi.org/10.1136/jech.2004.021204

The type of data for the explanatory variables and how it was treated in the regression analysis to be clearly stated. 

We assume the reviewer refers to whether data was categorical or continuous? We added the following sentence in the Results section (see Line 295 -> beginning of results on models): “Explanatory variables age, sleep, physical activity, stress, stressful life events, weight satisfaction, social support, and PCA scores were entered in the model as continuous variables. Gender and ethnicity were entered as categorical variables.”

If your comment refers to something else, please let us know and kindly clarify.

A statement on statistical test assumptions fulfillment would be useful.

The sentence “Assumptions for linear models were met” is already included in the manuscript, please see Line 231.

We added the following details (see Line 232): “Assumptions were checked as follows: Independence of cases was given due to the study design (each observation exists only once, is not paired with an observation in another group nor is it influenced by another observation). Collinearity was assessed through VIF values (largest VIF should be <10; average VIF should not be substantially >1) and tolerance statistics (which should be >0.2; Field, 2013). Normality was assessed through the normal probability plots of the residuals. Homoscedasticity and linearity were checked through residuals vs. fitted plots.”

Table 1, the number of missing data for individual variable to be displayed. 

Done, please see revised manuscript.

Table 2, for component 2, the reason(s) for the two negative values to be explained.

Principal Component Analysis (PCA) is based on correlations among variables. If variables in a component are positively correlated with each other, the loadings will be positive. If there are negative correlations, some of the loadings will be negative. In this example, you would expect that plant-based meat and dairy alternatives are inversely correlated with animal-based foods.

We added the following sentences to the manuscript in Line 267: “Two items with cross-loadings were observed. The items on plant-based alternatives to meat and dairy milk had positive loadings on the first component (plant foods) and negative cross-loadings on the second component, which are the animal-based foods.”

Table 3, total N to be stated. Title too short.

Done, please see revised manuscript.

Table 4, for item categories, the categories name to begin with capital letter.

Done, please see revised manuscript. To make sure all tables are consistent, we made the same changes in Table 1.

Table 5, B to be written as Unstandardized B.

Done, please see revised manuscript.

Line 281, for anxiety 65.1% but in Table 4 it showed 65.2%.

Done, please see revised manuscript. 

Line 297 – 303, results to be presented in table form.

The results described in this section refer to the unadjusted associations between the main explanatory variable and the outcomes of interest. These unadjusted correlations do not remain significant once confounding variables are taken into consideration. In this study, we particularly aimed to highlight the importance of consideration of confounding variables as unadjusted effects have been overrepresented in past studies. This is why we originally omitted presentation of the unadjusted correlations in table form. However, we now added this table, please see what is now Table 5.

Decimal point for percentages in the text in the results section to be standardized, i.e. at least 1 decimal point.

Done, please see revised manuscript. As is customary, we kept the 2 and 3 decimal points for the results of Beta and p-values, respectively.

Discussion

Line 401, for the sentence ‘relatively small sample size and associated lack of power’ figures to be provided to support the statement.

We realize that this sentence was misleading. We rephrased to the following in Line 406: “The relatively small sample size and the potential lack of power…” and added this sentence in Line 408: “This could be mitigated in future studies with larger sample sizes.”

Not all references are conformed to the journal format.

We double-checked this before resubmission and hope to not have missed any errors.

---

## [Editor Report · Decision Letter 1]

14 May 2021

The association between diet and mental health and wellbeing in young adults within a biopsychosocial framework

PONE-D-20-39899R1

Dear Dr. Rossa-Roccor,

We’re pleased to inform you that your manuscript has been judged scientifically suitable for publication and will be formally accepted for publication once it meets all outstanding technical requirements.

Kind regards,

Sıdıka Bulduk, Prof. Dr.

Academic Editor

PLOS ONE
---

## [Editor Report · Acceptance letter]

24 May 2021

PONE-D-20-39899R1 

The association between diet and mental health and wellbeing in young adults within a biopsychosocial framework 

Dear Dr. Rossa-Roccor:

I'm pleased to inform you that your manuscript has been deemed suitable for publication in PLOS ONE. Congratulations! Your manuscript is now with our production department. 

Kind regards, 

on behalf of

Dr. Sıdıka Bulduk 

Academic Editor

PLOS ONE